# Dextran-Chitosan Composites: Antioxidant and Anti-Inflammatory Properties

**DOI:** 10.3390/polym15091980

**Published:** 2023-04-22

**Authors:** Anca Roxana Petrovici, Narcis Anghel, Maria Valentina Dinu, Iuliana Spiridon

**Affiliations:** “Petru Poni” Institute of Macromolecular Chemistry, 41A Grigore Ghica Voda Alley, 700487 Iasi, Romania; petrovici.anca@icmpp.ro (A.R.P.); vdinu@icmpp.ro (M.V.D.)

**Keywords:** chitosan, dextran, biological properties

## Abstract

This study presents the development of new formulations consisting of dextran (Dex) and chitosan (Ch) matrices, with fillings such as chitosan stearate (MCh), citric acid, salicylic acid, or ginger extract. These materials were characterized using Fourier-Transform Infrared Spectroscopy (FTIR), Scanning Electron Microscopy (SEM), and mechanical tests, and evaluated for antioxidant properties, including scavenging activities, metal chelation, and ferric ion reducing power, as well as anti-inflammatory properties, measuring the binding affinity between serum albumin and the bioactive substances, which can influence their bioavailability, transport, and overall anti-inflammatory effect. Compounds in ginger such as 6-gingerol reduce inflammation by inhibiting the production of inflammatory substances, such as prostaglandin, cytokines, interleukin-1β, and pro-inflammatory transcription factor (NF-κB) and, alongside citric and salicylic acids, combat oxidative stress, stabilizes cell membranes, and promote membrane fluidity, thereby preserving membrane integrity and function. Incorporating chitosan stearate in chitosan:dextran samples created a dense, stiff film with an elastic modulus approximately seventeen times higher than for the chitosan:dextran matrix. The Dex:Ch:MCh sample exhibited low compressibility at 48.74 ± 1.64 kPa, whereas the Dex:Ch:MCh:citric acid:salicylic acid composite had a compact network, allowing for 70.61 ± 3.9% compression at 109.30 kPa. The lipid peroxidation inhibitory assay revealed that Dex:Ch:MCh:citric acid had the highest inhibition value with 83 ± 0.577% at 24 h. The study highlights that adding active substances like ginger extract and citric acid to Dex:Ch composites enhances antioxidant properties, while modified chitosan improves mechanical properties. These composites may have potential medical applications in repairing cell membranes and regulating antioxidant enzyme activities.

## 1. Introduction

Polysaccharides are a class of carbohydrates that consist of long chains of monosaccharide units [1]. These molecules are ubiquitous in nature and play a critical role in a wide range of biological processes. Polysaccharides are found in a variety of natural sources such as plants, animals, fungi, and bacteria [2]. There has been a growing interest in the use of natural polysaccharides as starting materials for the development of composite materials for medicine and cosmetics [3].

Natural polysaccharides are a renewable resource that can be obtained from a variety of sources such as seaweed, chitin, starch, and cellulose. These materials are abundant, and their production is not dependent on fossil fuels or other non-renewable resources. Furthermore, polysaccharides are biocompatible [4], meaning that they do not cause adverse reactions when introduced to biological systems. This makes them an attractive starting material for medical and cosmetic applications.

One example of the use of natural polysaccharides in medicine is the development of drug delivery systems [5,6]. Polysaccharides can be modified to create micro- or nanoparticles that can be loaded with drugs and targeted to specific cells or tissues. These delivery systems can improve drug efficacy and reduce side effects. Additionally, because natural polysaccharides are biodegradable, they can be broken down and eliminated from the body without causing harm.

In the field of cosmetics, natural polysaccharides have also been used as starting materials for the development of composite materials. Polysaccharides such as chitin, dextran, xanthan, alginate, and cellulose can be used to create films [7], gels, and other materials that can improve the texture and stability of cosmetic products.

Overall, the use of natural polysaccharides as starting materials for the development of composite materials for medicine and cosmetics is an exciting area of research. As research in this field continues, it is likely that even more innovative uses for natural polysaccharides will be seen in the future.

Due to its antibacterial and antifungal activity, biocompatibility and biodegradability, as well as its non-toxicity, chitosan is used in numerous applications. Chitosan is the second most abundant biological resource [8]. It is a hydrophilic linear polysaccharide obtained from marine invertebrates, composed of repeating units of d-glucosamine and *N*-acetylglucosamine. The hydroxyl at C_3_ and C_6_ and primary amine at C_2_ in every sequence gives it a high potential for being chemically transformed. Due to their biological and good film-forming properties, chitosan and its derivatives have become an attractive alternative to synthetic plastic polymers, with chitosan-based materials having various environmental, packaging or medical applications [9]. 

Chitosan has been used in medicine for a variety of purposes, including drug delivery, wound healing, and tissue engineering. Chitosan can be formulated into various drug delivery systems, such as nanoparticles, microparticles, and hydrogels, to improve drug solubility, stability, and bioavailability. For example, chitosan nanoparticles have been developed for the delivery of anticancer drugs, such as doxorubicin and paclitaxel, in order to improve their therapeutic efficacy and reduce their toxicity [10,11].

Chitosan has also been shown to promote wound healing by enhancing cell proliferation, migration, and differentiation. Chitosan-based dressings have been developed for the treatment of chronic wounds, burns, and ulcers, which can improve wound healing outcomes and reduce the risk of infection [12,13].

In addition, chitosan has been used in tissue engineering to fabricate scaffolds for the regeneration of various tissues, such as bone, cartilage, and skin. Chitosan-based scaffolds have been shown to support cell attachment, proliferation, and differentiation, and can promote tissue regeneration in vivo [14,15].

Chitosan and its derivatives have also been used in cosmetics due to their various skin benefits, such as moisturizing, anti-aging, and anti-acne effects. Chitosan can form a film on the skin, which can reduce water loss and improve skin hydration. Chitosan has also been shown to stimulate collagen synthesis and inhibit the activity of elastase, which can improve skin elasticity and reduce the appearance of wrinkles [16].

Moreover, chitosan has antimicrobial properties and can inhibit the growth of various bacteria and fungi, which can be useful in the treatment of acne and other skin infections. Chitosan-based formulations, such as creams and gels, have been developed for the treatment of acne, which can reduce the severity of acne lesions and improve skin texture [17,18].

Dextran is a polysaccharide produced by extracellular enzymes of some lactic acid bacteria [19,20]. Dextran produced by *Weissella* strains is of interest due to its linear backbone with only 3–4% α-(1→3) branching [21]. It has been used in pharmaceutical applications due to its prebiotic effect and its potential role in pathogen inhibition [22,23]. 

Dextran is extensively utilized in medicine as a blood plasma expander due to its capacity to boost blood volume and enhance blood flow in patients. Its large molecular weight and excellent water solubility contribute to its efficacy as a volume expander, leading to improved cardiac output and oxygen delivery to tissues. Additionally, dextran serves as a drug delivery agent, refining the pharmacokinetic profile of therapeutic drugs by prolonging their circulation time in the bloodstream and directing them to specific organs or tissues. For instance, dextran conjugated with the anticancer drug doxorubicin has demonstrated promising outcomes in treating various cancers, such as breast and ovarian cancer, by targeting tumor cells and minimizing side effects on healthy cells [24,25].

In addition to dextran, some of its derivatives have shown potential applications in medicine. For instance, dextran sulfate has been used as an anticoagulant in hemodialysis due to its ability to inhibit the activation of coagulation factors. Dextran sulfate has also been used as a therapeutic agent for HIV/AIDS, where it has been shown to inhibit the virus’s entry into host cells by binding to viral proteins [26].

Apart from medicine, dextran and its derivatives have been used in cosmetics as emulsifiers, thickeners, and moisturizers due to their ability to form stable gels and films. Dextran derivatives, such as carboxymethyl dextran, have been used in cosmetics as a moisturizer to improve skin hydration and prevent water loss [27]. Additionally, some studies propose that dextran-based nanoparticles may serve as drug delivery systems in cosmetics for skin-targeted treatments. For instance, curcumin-loaded dextran nanoparticles, which possess strong antioxidant and anti-inflammatory properties, have demonstrated penetration into the skin’s outer layer and reduction of skin inflammation. This indicates potential applications in addressing skin conditions such as psoriasis and eczema [28,29].

Dextran and its derivatives have shown significant potential in medicine and cosmetics. The unique physicochemical and biological properties of dextran make it a valuable material for various applications, including blood plasma expansion, drug delivery, as an anticoagulant, and for antiviral therapy. Additionally, dextran derivatives have shown potential as moisturizers, emulsifiers, and drug delivery systems in cosmetics. Future research could lead to the development of novel dextran-based materials with improved properties for various applications.

Natural compounds have a have drawn wide attention for the prevention and treatment of various diseases. Thus, ginger (*Zinziber officinale Rosc*.) rhizome is used as a condiment and spice in Asian countries and has become a significant product in the global food market. It also acts as a remedy for numerous diseases. Ginger’s biologically active compounds offer anti-inflammatory, antibacterial [30] and anticancer properties [31]. A recent review demonstrated antiviral, anti-inflammatory, antioxidative and immunomodulatory effects of ginger against SARS-CoV-2 infection. The main ginger polyphenols are gingerols, shogaols, and catechins, 6-gingerol being the most important bioactive compound with pharmacological effects [32]. 

Because natural polymers are inexpensive and can be combined to create novel materials with the appropriate qualities for a variety of applications, there is considerable interest in this topic from both the industrial and academic sectors. When collagen and epidermal growth factor were added into oxidized dextran and carboxyethyl chitosan precursors, a bioactive hydrogel for diabetic wound healing was prepared [33]. Chaiyasan et al. [34] prepared biocompatible mucoadhesive chitosan–dextran sulfate nanoparticles for lutein delivery to the ocular surface, while other authors [35] prepared, by a simple co-acervation method, curcumin-loaded dextran sulphate-chitosan nanoparticles as a promising formulation in cancer therapy. Further, a delivery system comprised of bovine serum albumin loaded in chitosan-dextran sulfate nanoparticles with application in bone regeneration processes [36] has been reported. It is worth mentioning the importance of material composition and the component ratio in the obtainment of such systems.

While there are no specific studies that have combined chitosan stearate, dextran, ginger extract, citric acid, and salicylic acid formulations, there are individual studies that have explored the use of these components in various applications. Thus, Ismail et al. [37] have developed chitosan-based wound dressings incorporating ginger extract and curcumin, which demonstrated enhanced wound healing properties. Guerrero et al. made chitosan microcapsules loaded with salicylic acid, which were prone to inhibit the mycelial growth of *Alternaria alternata*, *Botrytis cinerea*, *Fusarium oxysporum* and *Geotrichum candidum* strains [38], and Hernández et al. manufactured chitosan-citric-acid based antimicrobial films which were shown to be active against external contamination by *Zygosaccharomyces bailii* [39].

Herein, we report the obtainment of new formulations based on Dextran (Dex) and chitosan (Ch) as matrix, and as fillings using chitosan stearate (MCh), citric acid (CA), salicylic acid (SA) or ginger extract (VE). Fourier-Transform Infrared Spectroscopy (FTIR), Scanning Electron Microscopy (SEM) and mechanical tests were performed in order to characterize all materials. The biological and anti-inflammatory properties were also evaluated. Our study points out the antioxidant and anti-inflammatory properties of these materials, which make them suitable candidates for use in cosmetic applications.

## 2. Materials and Methods

### 2.1. Materials

Potassium thiocyanate, 1-Buthyl-3-methyl-imidazolium, chloride, ethyl-3-methyl-imidazolium tetrafluoroborate, DPPH (2,2-Diphenyl-1-picrylhydrazyl hydrate, 95%), ferrous chloride tetrahydrate, ferric chloride tetrahydrate, ferrozine, potassium ferricyanide, bovine albumin, Trolox (6-hydroxy-2,5,7,8-tetramethylchromane-2-carboxylic acid, 97%), l-ascorbic acid, linoleic acid, stearic anhydride, Tween-20, ABTS (2,2′-azino-bis(3-ethylbenzothiazoline-6-sulphonic acid) diammonium salt, pyrogallol, phenanthroline, trichloroacetic acid, citric acid, salicylic acid, and chitosan (Ch) with medium molecular weight were purchased from Sigma-Aldrich Chemie GmbH, Germany. 

Dextran, with a molecular mass of 6.6 × 10^5^ Da, was obtained by fermentation using a bacteria strain of *Weissella Confusa* isolated from Romanian commercial yoghurt in the laboratories of the Centre of Advanced Research in Bionanoconjugates and Biopolymers (IntelCentru) of the “Petru Poni” Institute of Macromolecular Chemistry, according to the procedure presented in [20].

Ginger extract (VE), having a total phenolic content (TPC) of 122.4 μg/mL gallic and 81.64 μg/mL flavonoids (expressed as rutin equivalents), was obtained by the method described by Ivane et al. [40]. Briefly, after being ground into a fine powder and put through a 20 mesh filter, some dried ginger was sonicated with ethanol for 15 min. The mixture was put into a centrifuge tube and centrifuged for 5 min at 3000× *g* to separate the supernatant. To produce a ginger ethanolic extract solution, this procedure was carried out twice. The ethanolic extract was next concentrated using a vacuum rotatory evaporator set at 50 °C. For later usage, the resulting concentrate was kept in storage at 4 °C. TPC was determined by Folin–Ciocalteu’s method [41].

### 2.2. Chitosan Modification

Total amounts of 7 g of 1-buthyl-3-methyl-imidazolium chloride, 4.6 g ethyl-3-methyl-imidazolium tetrafluoroborate, and 0.5 g chitosan were mixed and kept at 100 °C. After 12 h, 1.6 g stearic anhydride was added and the mixture was stirred. When the temperature reached room temperature, 0.2 g lipase was added. After 24 h, the mixture was treated with ethanol. The modified chitosan was separated by centrifugation (4000 rpm, 10 min), washed 3 times with acetone, and dried in a vacuum oven at 60 °C for 4 h.

### 2.3. Preparation of Films

A 1% chitosan (Ch) solution was prepared by dissolution in 0.5% acetic acid. Dextran (Dex) was dissolved in water and different formulations comprising modified chitosan (MCh), citric acid (CA), salicylic acid (SA), and ginger extract (VE) were prepared, according to Table 1.

### 2.4. Characterization of Films

#### 2.4.1. Scanning Electronic Microscopy (SEM)

SEM was carried out using a Quanta 200 Scanning Electron Microscope (FEI Company, Hillsboro, OR, USA) using a low vacuum secondary electron detector and an accelerating voltage of 25.0 kV. Samples were fixed onto aluminum stubs by carbon adhesive disks.

#### 2.4.2. Fourier-Transform Infrared Spectroscopy (FTIR) 

FTIR spectra of the materials were recorded using a Vertex 70 FTIR spectrometer from Brüker, equipped with an ATR (Attenuated Total Reflectance) device (ZnSe crystal), at a 45 angle of incidence. The spectra were recorded by accumulation of 32 scans in the range of 400–4000 cm^−1^ at room temperature with a resolution of 2 cm^−1^.

#### 2.4.3. Mechanical Characterization of Biomaterials

A Shimadzu Testing Equipment (EZ-LX/EZ-SX Series, Kyoto, Japan) was used to analyze the mechanical properties of all samples. The uniaxial compression measurements were performed on samples, shaped as plates with a thickness, width and height of about 12 mm, 16 mm and 5 mm, respectively. The stress-strain profiles were registered using a force of 20 N and a crosshead speed of 1 mm × min^−1^. An initial force of 0.1 N was set out to apply on each sample before each compression measurement to guarantee a complete contact between the biomaterial surface and the compression plate of the analyzer. The compressive stress (*σ*, kPa) was calculated as the normal force (*F*, N) acting perpendicular to the area of the plate (*A*, m^2^), while the strain (ε) was evaluated as the ratio between the change in length (Δ*l*, m) and initial length (*l_o_*, m). The compressive elastic modulus (*G*, kPa) was determined as the slope of the initial linear portion in the stress-strain profiles in agreement with the procedure previously described for other porous biomaterials [42,43].

### 2.5. DPPH Radical Scavenging Assay

In order to establish DPPH radical scavenging, a modified procedure, as published by Kitrytė et al. [44], was used as follows: an amount of 10 mg material was suspended into 500 μL DDW (double-distilled water) at the same time as the active substances used for loading the composites and mixed with 1000 μL DPPH radical, with 89.7 μmol/L concentrations in methanol, then left 2 h in the dark to complete the reaction. The DDW was used as a blank and the absorbance was measured in triplicate at 517 nm, and expressed as Trolox equivalents (mg TE/g samples), 0–50 μmol/L in methanol. 

The DPPH radical Inhibition percentage was calculated using Equation (1):(1)$$Inhibition=\left(1-\frac{A_{s}}{A_{0}}\right)\times 100,\;\%$$

*A_S_* is the sample absorbance; *A*_0_ is the blank absorbance. 

### 2.6. ABTS Radical Scavenging Assay

Radical scavenging properties were evaluated by mixing 0.3 mL of each sample in DDW and DDW as negative control, with 1.2 mL of ABTS reagent which was prepared 16 h before as follows: 350 μL ABTS diammonium salt (7.4 mmol/L) were mixed with 350 μL potassium persulfate (2.6 mmol/L), kept in the dark at room temperature in order to complete the radical generation, then diluted 1:50 with 95% ethanol. The mixtures were then incubated for 6 min and the absorbance was measured at 734 nm [44] and expressed as Trolox equivalents (mg TE/g samples) calculated by means of dose–response curves for Trolox (0–50 μmol/L in methanol). The ABTS inhibition percentage of the samples was calculated using Equation (1).

### 2.7. Superoxide Anion Radical (O_2_^−•^) Scavenging Activity

The studied samples’ superoxide anion radical scavenging activity was determined as follows: 10 mg of each sample were suspended in 40 μL DDW, distilled H_2_O as a control, mixed with 1.8 mL Tris-HCl (0.05 M, pH 8) then left to rest for 20 min at 25 °C. After this time, 160 μL of pyrogallol (25 mM) was added and the mixture was kept for another 5 min at 25 °C, then, 10 μL HCl (8 M) was added and at 325 nm the absorbance was measured. The radical scavenging percentage was calculated using Equation (1) [45]. 

### 2.8. Hydroxyl Radical (HO^−•^) Scavenging Ability

The hydroxyl radical scavenging abilities were determined by suspending 10 mg of each composite sample, L-ascorbic acid and distillated H_2_O as a control in 400 μL DDW and mixed vigorously with 400 μL phosphate-buffered saline (PBS, 0.2 M, pH 7.4) and 400 μL *O*-phenanthroline (2.5 mM). Then, 400 μL FeSO_4_·7H_2_O (2.5 mM) and 400 μL H_2_O_2_ were added and the mixture was incubated 1 h at 37 °C, after which the absorbance was measured at 536 nm. The scavenging percentage of HO^−*•*^ was calculated using Equation (2) [45]: (2)$$Scavenging\;rate=\left(\frac{A_{s}-A_{1}}{A_{0}-A_{1}}\right)\times 100,\;\%$$
where *A_s_* is the absorbance of the sample; *A*_0_ is the absorbance of distilled water in reaction; and *A*_1_ is the absorbance of hydrogen peroxide in reaction.

### 2.9. Lipid Peroxidation Inhibitory Assay

The composites’ inhibitory activities were measured as follows: each sample (10 mg suspended in 100 μL DDW), l-ascorbic acid and DDW as a control was mixed with 900 μL phosphate buffer (dipotassium hydrogen phosphate in distilled water 0.2 M at pH 7) and 1000 μL linoleic acid emulsion. The fresh emulsion was prepared by mixing 155 μL linoleic acid and 175 μg Tween-20 in 50 mL phosphate buffer 0.2 M. The resulting mixture was incubated at 37 °C and after 1 and 24 h, 50 μL of this solution was taken and mixed with 1.85 mL ethanol and 50 μL FeCl_2_·4H_2_O solution (20 mM in 3.5% HCl). The resulting solution was mixed thoroughly and 50 μL potassium thiocyanate (30% in distilled water) was added. The inhibitory effect was calculated using Equation (1) after recording the absorbance at 500 nm. 

### 2.10. Ferrous Ions’ (Fe^2+^) Chelating Activity

The composites’ chelating activity was compared to l-ascorbic acid (as a common antioxidant reference) and was estimated as follows: 200 μL of DDW samples suspension, l-ascorbic acid, active substances and DDW as a control and 100 μL ferrous chloride tetrahydrate (2 mM) was mixed. Over the mix, 200 μL ferrozine (2 mM) and 1.5 mL ethanol were added, which was then vigorously shaken and incubated at 10 min room temperature. The inhibition percentage of Fe^2+^–ferrozine complex was calculated after the absorbances were read at 562 nm by using Equation (1). 

### 2.11. Ferric Ions (Fe^3+^) Reducing Antioxidant Power (FRAP) Assay

FRAP assay was performed using a previously published protocol [45]. For the testing, we used all the studied samples, L-ascorbic acid and DDW in a volume of 50 μL and mixed with 650 μL potassium ferricyanide (1%) and 650 μL sodium phosphate buffer (0.2 M, pH 6.6) then incubated at 50 °C for 20 min. After, 650 μL trichloroacetic acid (10%) was added to the mixture and in a separate tube 180 μL ferric chloride (FeCl_3_, 0.1%), 910 μL DDW and 910 μL of the mixture solution was vortexed. The reducing power (%) was expressed as the ratio between the absorbance of the sample and the higher absorbance measured at 700 nm.

### 2.12. Anti-Inflammatory Properties

To assess the anti-inflammatory properties of the materials, a modified version of the protein denaturation method [46] was employed. The reaction mixture was comprised of 100 mg of the individual materials (Dex:Ch:CA:SA, Dex:Ch:CA:VA, Dex:Ch:MCh:CA:SA and Dex:Ch:MCh:CA:VA) and 2 mL of 0.1% bovine albumin solution in 5 mL of saline phosphate buffer (PBS) with a pH of 6.4. The mixture was incubated in a water bath at 37 °C for 15 min and then heated to 70 °C for 5 min. Following cooling, the absorbance was measured at 660 nm using PBS solution as the blank, while the solution of bovine albumin was utilized as the control. Each experiment was conducted in triplicate, and Equation (3) was used to calculate the inhibition percentage of the anti-inflammatory activity
(3)$$\%\;inhibition=100\times \left(1-\frac{A_{s}}{A_{c}}\right)$$
where *A_s_* is absorption of the sample and *A_c_* is absorption of the control. 

## 3. Results and Discussion

### 3.1. FTIR Analysis

The esterification of chitosan with stearic acid was investigated using the FTIR to confirm that the reaction took place.

In Figure 1A, the infrared spectrum of chitosan displays several absorption bands, including a strong band within the range of 3300–3600 cm^−1^, which corresponds to the stretching of N–H and O–H as well as intramolecular hydrogen bonds. Furthermore, the bands observed at approximately 2918 and 2855 cm^−1^ represent the symmetric and asymmetric stretching of C–H, respectively, and are typical characteristics of polysaccharides like xylan, xanthan, alginate, and carrageenan [47,48,49]. The presence of residual N–acetyl groups is confirmed by the absorption bands at approximately 1661 cm^−1^ (C=O stretching of amide I) and 1319 cm^−1^ (C–N stretching of amide III). The N–H bending of the primary amine corresponds to a band at 1597 cm^−1^ [50]. 

Moreover, the presence of CH_2_ bending and CH_3_ symmetrical deformations is validated by the presence of bands at approximately 1381 and 1425 cm^−1^, respectively. Additionally, the band at 1155 cm^−1^ is attributed to the asymmetric stretching of the C–O–C bridge, while the bands at 1032 and 1076 cm^−1^ correspond to C–O stretching. It is noteworthy that all of these bands conform to the spectra of chitosan presented in previous studies [51,52].

Figure 1B evidences the hydroxyl groups at 3433 cm^−1^ and an intensification, due to the presence of hydrocarbonate chain of stearic acid, of the stretching vibrations of the –CH_2_– groups at 2918 cm^−1^ for MCh. Additionally, an increase of the absorption for the band at 1737 cm^−1^, characteristic for the carbonyl group, was recorded. These findings confirm the introduction of new ester groups into the structure of chitosan.

### 3.2. Mechanical Properties

The uniaxial compressive tests were carried out to establish the influence of chemical composition on the mechanical features of the composite biomaterials. The compressive stress–strain (σ-ε) profiles depicted in Figure 2A for the dextran/chitosan (Dex:Ch)-based composites indicate a typical elastic behavior characteristic for macroporous materials. The Chi, Dex:Ch, Dex:Ch:CA, Dex:Ch:CA:SA or Dex:Ch:CA:VE biomaterials can sustain compression values beyond 60% without any failure or deformation of the gel network (Figure 2A). 

It should be pointed out that, by the addition of the third component (citric acid) or the fourth component (salicylic acid or vegetal extract) the compressive strength increased, indicating an increase of the network rigidity. For instance, the Dex:Ch:CA:SA and Dex:Ch:CA:VE composites sustained 70.86 ± 2.7%, and 64.75 ± 2.16% compression, respectively, at a compressive nominal stress of 74.35 kPa and 70.15 kPa, while the Dex:Ch:AC, Dex:Ch, and Ch sustained 73.04 ± 3.14%, 68.69 ± 4.7%, and 85.72 ± 3.4% compression, respectively, at a compressive nominal stress of 62.94 kPa, 37.98 kPa, and 52.25 kPa (Figure 2C). The compressive moduli of Ch, Dex:Ch, Dex:Ch:CA, Dex:Ch:CA:SA or Dex:Ch:CA:VE biomaterials, calculated as the slope of the initial linear portion in the stress-strain profiles (Figure 2B), increased with the addition of the third component (citric acid) or the fourth component (salicylic acid or vegetal extract). The Dex:Ch:CA, Dex:Ch::A:SA and Dex:Ch:CA:VE composites exhibited an elastic modulus about five, fifteen, and fourteen-times higher, respectively, than that of the starting network (Ch) (Figure 2D). The improvement of the mechanical performances of the biomaterials containing three or four components in comparison to those with one network could be associated with the changes in the sample’s morphology as well as the decrease in the pore sizes with the increase of the components number in the matrices (Table 1). These results are in accordance with the data reported on Ch/dextrin sponges [43] and Ch/agarose blends [53].

On the other hand, by increasing the polymer content through embedment of a modified chitosan (MCh) within Dex:Ch sample, quite dense and stiff composite networks were obtained (Figure 3). The Dex:Ch:MCh sample exhibited an elastic modulus about seventeen times higher than that of the Dex:Ch network (see Figure 2 and Figure 3).

Moreover, from comparing Dex:Ch:CA and Dex:Ch:MCh:CA samples it can be concluded that both composites were mechanically stable but that the latter sustained only 37.97 ± 6.1% compression at a compressive nominal stress of 48.74 kPa, while the former sustained 73.04 ± 3.14% compression at a compressive nominal stress of 62.94 kPa (Figure 3A,C). Similar results were previously shown for polyelectrolyte complex cryogels based on Ch and carboxymethylcellulose [54]. 

The introduction of the fifth component, i.e., salicylic acid, within the Dex:Ch:MCh:CA composite led to an enhancement of the compressive strength of the biomaterial (sample Dex:Ch:MCh:CA:SA; Figure 3B). The composite Dex:Ch:MCh:CA:SA sustained 70.61 ± 3.9% compression at a compressive nominal stress of 109.30 kPa, which can be attributed to its compact network with isolated pores and thicker pore walls (see SEM micrographs—Figure 4, Table 2). In addition, the toughness of porous materials is affected by the average pore sizes and the hydrogel networks containing well-defined interconnected structures exhibited an enhancement of the elastic moduli [55,56].

Therefore, the improvement of the compressive elastic moduli for Dex:Ch:MCh:CA and Dex:Ch:MCh:CA:VE composites (Figure 3D) can be associated with the well-defined morphology of honeycomb- or lamellar-like pores observed for these biomaterials (see SEM micrographs, Figure 4). 

### 3.3. Biological Properties of Materials

It is well known that the human metabolism generates many free radicals inside the body, particularly reactive oxygen species (ROS) such as H_2_O_2_, superoxide anion and hydroxyl radicals. These ROS produce oxidative damage to cellular systems by attaching to DNA, proteins, lipids and by inducing inflammation [57], which can further generate different pathologies. Antioxidant compounds have positive effects due to their properties to scavenge free radicals. If new drug delivery systems are developed, it is very useful if their antioxidant properties are determined in order to delimitate the therapeutic approaches. 

DPPH and ABTS assays are commonly used methods for determining the ability of a system to reduce radicals [58]. As can be seen in Figure 5A,B, our compounds show the ability to reduce these radicals. 

Chitosan exhibits antioxidant capabilities against DPPH radicals with a value of 24.6 ± 1.73% and ABTS radicals with a value of 84.1 ± 1.42%, respectively. These values are strongly reduced for the Dex:Ch matrix. Addition of MCh into the Dex:Ch matrix increased both ABTS and DPPH. The higher scavenging capacity of MCh could be related to more linkages between matrix components involved in breaking the free radical chain reactions, which could function as hydrogen donor to DPPH and ABTS radicals. 

As can be seen in Figure 5A, the addition of modified chitosan (MCh) in the composite formulation comprising ginger extract (VE) causes an increase of DPPH activity from 15% to 50%. Our data are in agreement with other authors’ results according to which the addition of polyphenolic compounds in Ch polymeric systems increases the Ch antioxidant activity [59]. 

The Dex:Ch:CA:VE system recorded the highest ABTS radical inhibition (86.37 ± 1.52%), which means that antioxidant compounds from ginger extract act as reducing agents and electron donors. 

The testing for superoxide anion and hydroxyl radical scavenging activity was performed to evidence if the studied materials have the capacity to scavenge radicals that are usually formed at inflammation sites. As a result of their accumulation at inflammation sites, they can cause irremediable damage to surrounding tissues that can lead to serious pathologies. 

The polymer matrix that does not contain active substances presented a low scavenging activity (Figure 5B). The addition of CA slowly increased values for this parameter. Interestingly, the simultaneous presence of CA and SA in the polymer system resulted in more than twice the activity (86.21 ± 1.247%) as compared to that recorded when only the mixture of active substances is tested (34.36 ± 0.471%) (Figure 5B). The presence of modified chitosan in the polymeric matrix was beneficial for superoxide anion and hydroxyl radical scavenging activity. The best scavenging activity values were recorded for systems containing vegetal extract. These results are correlated well with anti-inflammatory activity, as shown in Figure 6.

Dex:Ch:MCh:CA:SA and Dex:Ch:MCh:CA:VE presented the highest anti-inflammatory activity, which can be attributed to MCh’s enhanced antioxidant potential, as demonstrated above via efficient scavenging for radical species.

Iron is a redox active metal which tends to undergo redox reactions and, as a result, generates many reactive oxygen species (ROS) which lead to lipid peroxidation, protein modification and other effects. Iron chelation will avoid its participation in redox reactions and prevent subsequent oxidative stress.

According to data presented in Figure 7A, the addition of CA, or CA and SA, resulted in a in a considerable increase in chelating activity. It seems that the presence of modified chitosan into the matrix negatively influenced the chelating capacity. The system Dex:Ch:CA:VE shows greater activity (32 units) even than VE (28 units) or CA:VE (20 units) tested separately. These results suggest a potential use in heavy metal decontamination.

The ferric ions’ reducing antioxidant power (FRAP) activity presented in Figure 7A confirm data reported by Zhang et al., which states that Ch presents a low value of FRAP activity [59]. The antioxidant activity of VE depends on the phenolic hydroxyl groups which interact with reactive oxygen species by hydrogen donating. It was found that the FRAP activity increased with the addition of ginger extracts into the polymeric matrix, the effect being more pronounced in the matrix without modified chitosan. A possible explanation could be the poor water solubility of MCh, which affects interactions between matrix components. 

To conclude, the presence of ginger extracts in the polymeric matrix increases iron chelation capacity, while the addition of modified chitosan into the matrix negatively influenced this parameter.

The lipid peroxidation inhibitory assay was carried out to see if the samples have the ability to neutralize the free radicals that lead to lipid peroxidation, thus slowing down the process of restoring cell membranes, as well as the destruction of intact cell membranes. Analyzing the data obtained at 24 h, we can say that Dex:Ch:MCh:CA presented the best inhibitory properties, with an inhibition value of 83.48 ± 0.577% (Figure 7B). Taking into account the latest research in the field, according to which Ch has cell membranes with reparative properties by regulating antioxidant enzyme activities, as well as decreased lipid peroxidation [60], the medical applications of this composite should be further tested with in vitro and in vivo tests.

The samples containing VE have an inhibition capacity of 92.17 ± 1.154% for Dex:Ch:CA:VE and 88.7 ± 0.577% for Dex:Ch:CA:MCh:VE. Their capacity is much lower than that of VE (79.1 ± 0.874%) when tested separately. We can conclude that the presence of VE in the polymeric matrix could prevent and/or attenuate the free-radical- induced peroxidation of cellular structures such as lipids, proteins, and DNA.

## 4. Conclusions

The data show the mechanical and antioxidant properties of composite materials made from Dex:Ch, modified chitosan (MCh), and active substances such as ginger extract (VE), citric acid (CA), and salicylic acid (SA). Dex:Ch:CA:SA and Dex:Ch:CA:VE composites sustained 70.86 ± 2.7% and 64.75 ± 2.16% compression, respectively, at compressive nominal stresses of 74.35 kPa and 70.15 kPa. Dex:Ch:AC, Dex:Ch, and Ch composites sustained 73.04 ± 3.14%, 68.69 ± 4.7%, and 85.72 ± 3.4% compression, respectively, at compressive nominal stresses of 62.94 kPa, 37.98 kPa, and 52.25 kPa. The addition of MCh to Dex:Ch resulted in a stiffer composite network with an elastic modulus about seventeen times higher than that of the Dex:Ch network. The Dex:Ch:MCh:CA:SA composite sustained 70.61 ± 3.9% compression at a compressive nominal stress of 109.30 kPa due to its compact network with isolated pores and thicker pore walls. The Dex:Ch:CA:VE system recorded the highest ABTS radical inhibition of 86.21 ± 1.247%. The addition of VE increased the DPPH antioxidant activity from 15% to 50%. The system Dex:Ch:MCh:CA presented the best inhibitory properties in the lipid peroxidation inhibitory assay, with an inhibition value of 83.48 ± 0.577% at 24 h. The data suggest that these composite materials have potential medical and cosmetic applications for repairing cell membranes and regulating antioxidant enzyme activities.

## Figures and Tables

**Figure 1 polymers-15-01980-f001:**
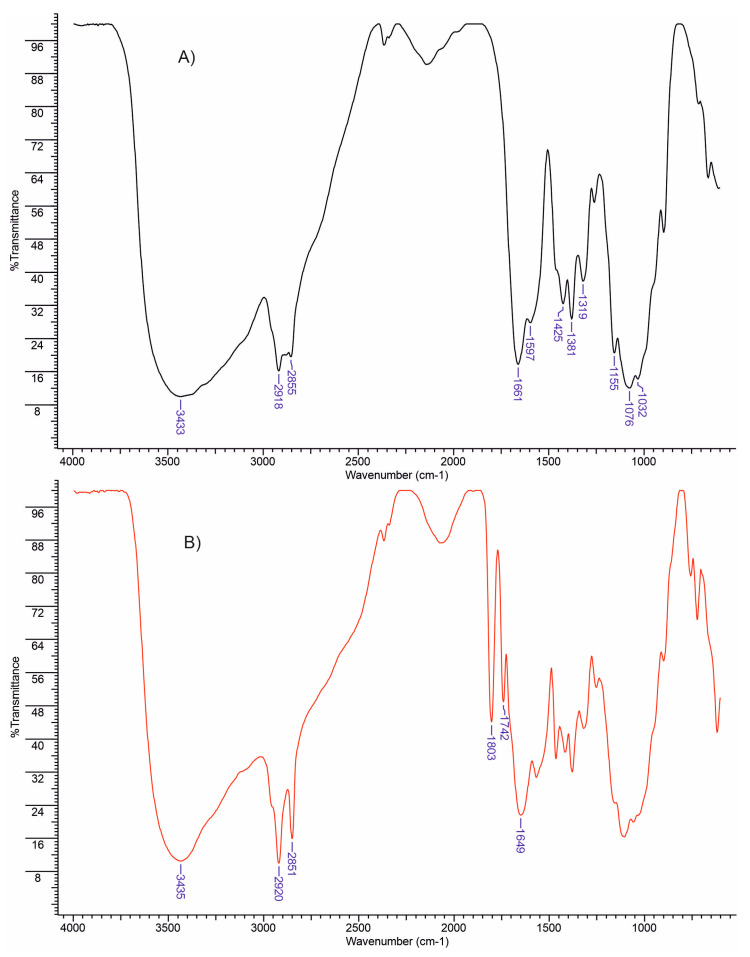
FTIR spectra of chitosan (**A**) and chitosan esterified with stearic anhydride (**B**).

**Figure 2 polymers-15-01980-f002:**
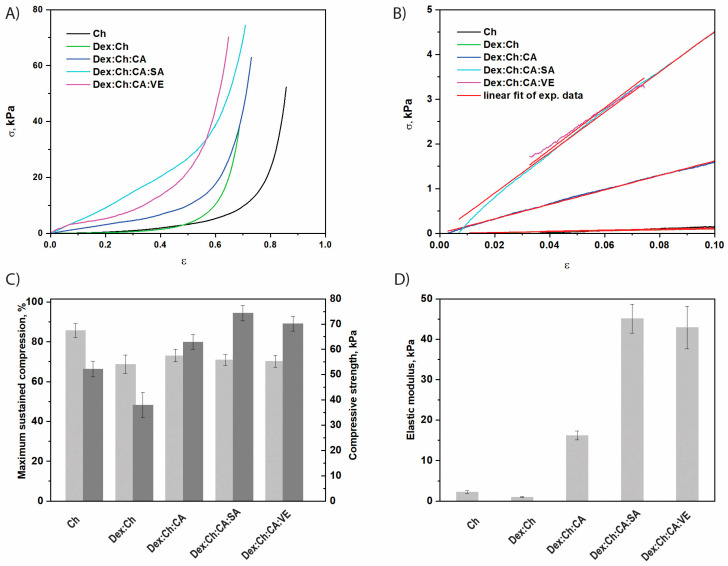
The mechanical properties of CS-based composite biomaterials: (**A**) Stress-strain profiles; (**B**) The linear dependence of stress-strain profiles used to evaluate the compression elastic moduli; (**C**) Maximum sustained compression (light grey) and compressive nominal stress (dark grey), *p* (probability value) = 0.043; (**D**) The values of compressive elastic modulus, *p* = 6.82 × 10^−5^. All values are presented as the average of at least three measurements ± standard deviations.

**Figure 3 polymers-15-01980-f003:**
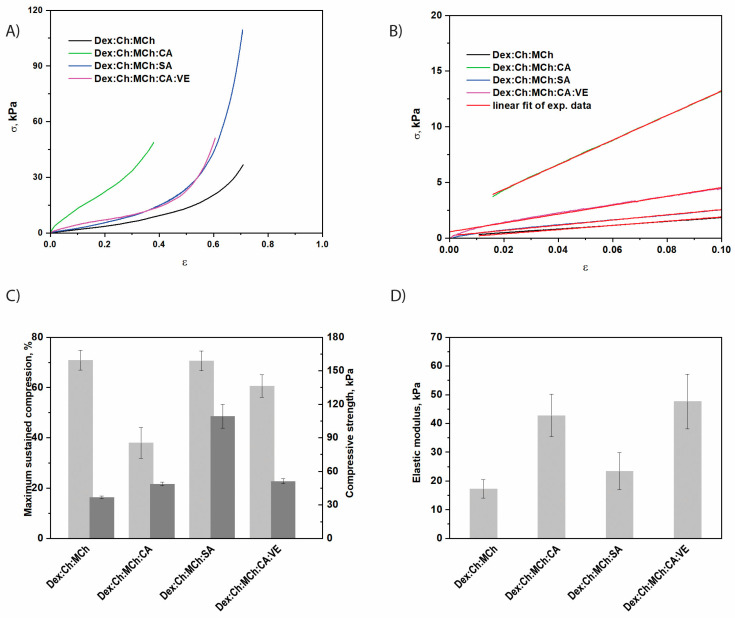
The mechanical properties of composite biomaterials containing modified chitosan (MCh) as the third network: (**A**) Stress-strain profiles; (**B**) The linear dependence of stress-strain profiles used to evaluate the compression elastic moduli; (**C**) Maximum sustained compression (light grey) and compressive nominal stress (dark grey), *p* = 0.048; (**D**) The values of compressive elastic modulus, *p* = 0.331. All values are presented as the average of at least three measurements ± standard deviations.

**Figure 4 polymers-15-01980-f004:**
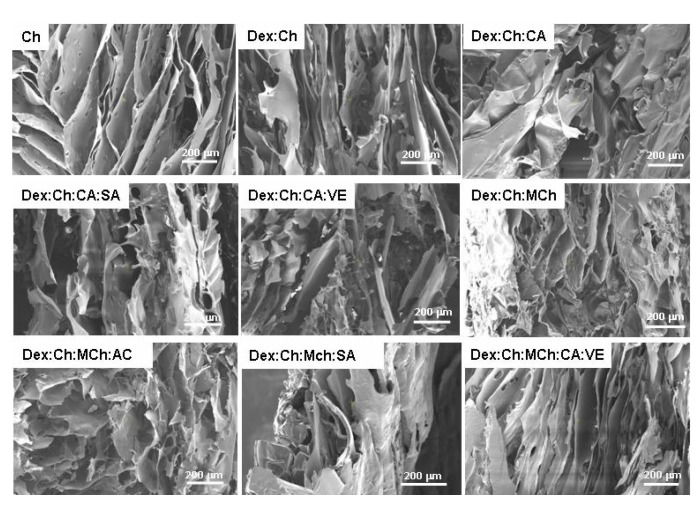
SEM micrographs of the investigated composite biomaterials as a function of chemical composition. Scaling bar 200 μm.

**Figure 5 polymers-15-01980-f005:**
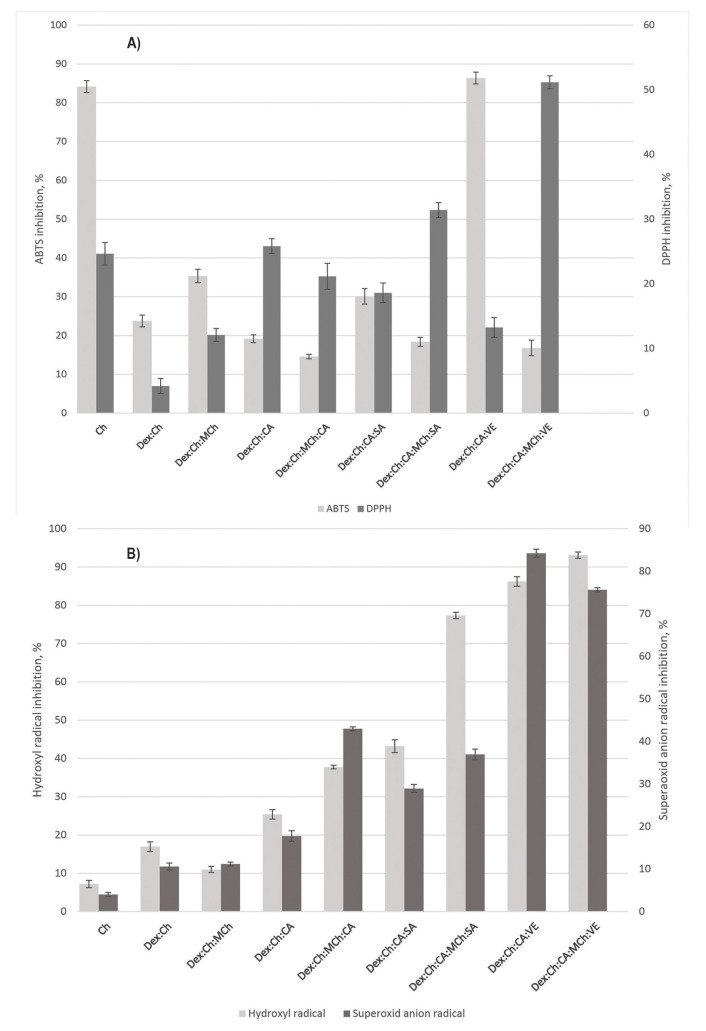
(**A**) Radical scavenging activity of tested materials: ATBS radical inhibition (light grey columns), *p* = 1.9 × 10^−22^; (**B**) DPPH radical inhibition (dark grey columns), *p* = 2.86 × 10^−17^; (**B**) Hydroxyl radical (light grey columns—*p* = 3 × 10^−26^) and superoxide anion radical (dark grey columns—*p* = 4.58 × 10^−27^) scavenging activity of tested materials.

**Figure 6 polymers-15-01980-f006:**
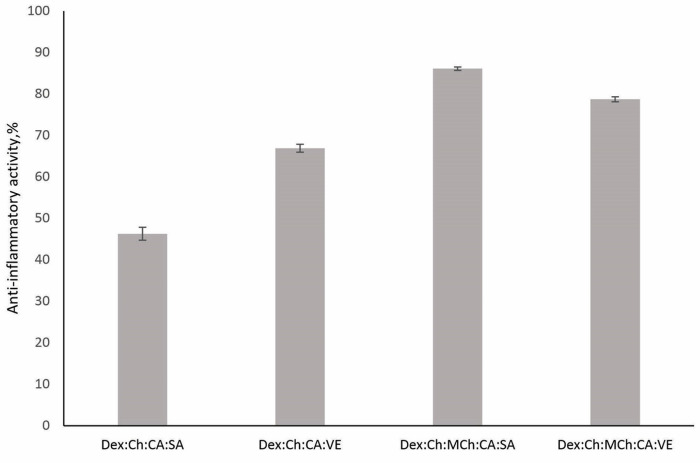
Anti-inflammatory activity of tested materials (*p* = 0.000994).

**Figure 7 polymers-15-01980-f007:**
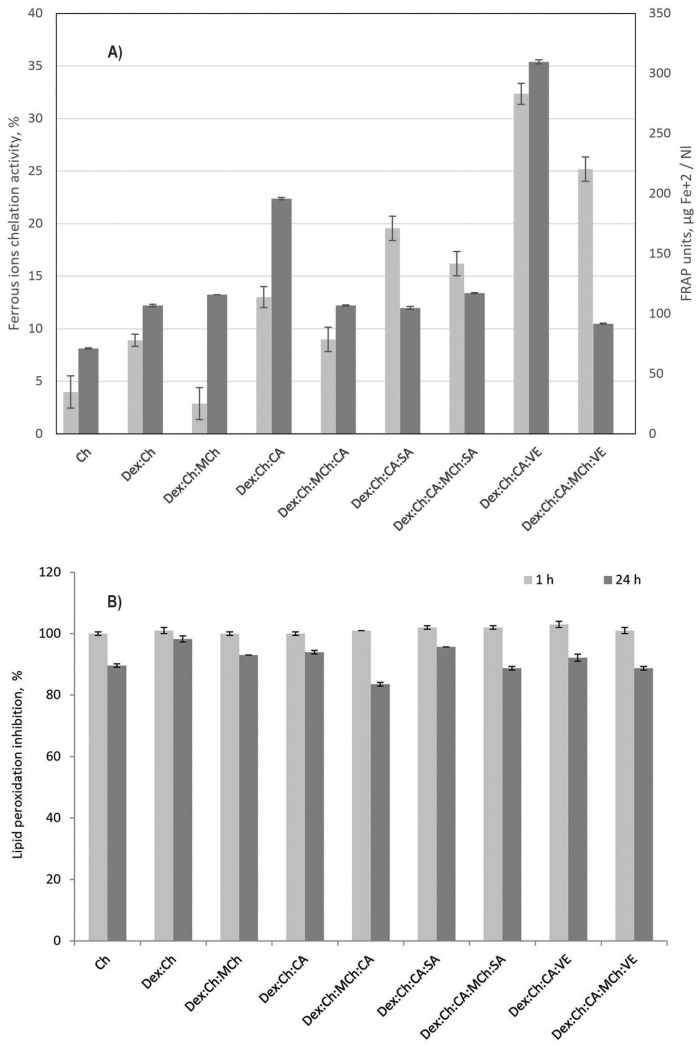
(**A**) Ferrous ions chelating activity (light grey columns—*p* = 3.48 × 10^−16^) and FRAP (dark grey columns—*p* = 4.92 × 10^−34^) of the studied materials; (**B**) Inhibition of lipid peroxidation by the studied materials (*p* = 2.32 × 10^−14^).

**Table 1 polymers-15-01980-t001:** Sample formulations.

Sample Code	Sample Formulation
Ch	chitosan
Dex:Ch	dextran:chitosan (1 g:1 g)
Dex:Ch:MCh	dextran:chitosan: modified chitosan (1 g:1 g:0.1 g)
Dex:Ch:CA	dextran:chitosan:citric acid(1 g:1 g:0.1 g)
Dex:Ch:MCh:CA	dextran:chitosan:modified chitosan: citric acid (1 g:1 g:0.1 g:0.1 g)
Dex:Ch:CA:SA	dextran:chitosan:citric acid: salicylic acid (1 g:1 g:0.1 g:0.1 g)
Dex:Ch:CA:MCh:SA	dextran:chitosan:citric acid: modified chitosan: salicylic acid (1 g:1 g:0.1 g:0.1 g:0.1 g)
Dex:Ch:CA:VE	dextran:chitosan: citric acid: ginger extract(1 g:1 g:0.1 g:0.1 g)
Dex:Ch:CA:MCh:VE	dextran:chitosan:citric acid:modified chitosan:ginger extract (1 g:1 g:0.1 g:0.1 g:0.1 g)

**Table 2 polymers-15-01980-t002:** Average pore sizes of composite biomaterials evaluated from SEM micrographs by ImageJ 1.41o software.

Sample Code	Average Pore Size, μm	Main Outcomes
Ch	77.61 ± 23.70	Lamellar morphology with interconnected pores
Dex:Ch	49.43 ± 8.80	Lamellar morphology with compact pore walls
Dex:Ch:CA	53.45 ± 9.75	Changed morphology: compact network with isolated pores
Dex:Ch:CA:SA	55.29 ± 8.95	Lamellar morphology with thicker pore walls
Dex:Ch:CA:VE	50.58 ± 6.91	Heterogeneous morphology with interconnected pores
Dex:Ch:MCh	65.18 ± 15.88	Heterogeneous morphology with elongated pores
Dex:Ch:MCh:CA	77.08 ± 14.90	Heterogeneous morphology with honeycomb-like pores
Dex:Ch:MCh:SA	44.72 ± 6.62	Changed morphology: compact network with isolated pores and thicker pore walls
Dex:Ch:MCh:CA:VE	35.17 ± 6.58	Lamellar morphology, thinner pore walls

## Data Availability

The data that support the findings of this study are available from the corresponding author upon reasonable request.

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
