# Peer review of "Dextran-Chitosan Composites: Antioxidant and Anti-Inflammatory Properties"

_polymers, 2023, doi:10.3390/polym15091980_

Round 1

Reviewer 1 Report

The MS needs more effort to consider further. 

Comments:

Briefly check the typo errors and grammar throughout. 

Abstract

 as fillings using chitosan stearate (MCh), citric acid, salicylic acid or ginger extract: Briefly explain the concept of this study?

The biological (hydroxyl, superoxid anion, and free radicals scavenging activity along with metal chelating properties and ferric ions reducing power): Though these are involved in biological processes, in most cases they are considered antioxidant properties. Now the word biological is misleading the paper. The title also confusing in this case, revise the title also.

 anti inflammatory properties were also evaluated: Better briefly explain them.

 modified chitosan within chitosan: Better to explain what is modified chitosan first. 

:dextran samples resulted in a dense and stiff: What was the form of your sample exactly? FIlms or scaffold or composite ...

Dextran:chitosan:chitosan stearate sample: Try to use some abbreviation for these samples.

 Dextran:chitosan:chitosan stearate sample exhibited low compression at a compressive nominal stress of 48.74 kPa. Dex:Ch:MCh:citric acid:salicylic acid composite sustained 70.61% compression at a compressive nominal stress of 109.30 kPa due to its compact network with isolated pores and thicker pore walls.: Simplify these sentences in one sentence.

 Explain the effect of citric acid, salicylic acid or ginger extract compared to modified chitosan in antioxidant  in abstract. 

Explain the anti-inflammatory effects of modified chitosan, citric acid, salicylic acid or ginger extract.

for repairing cell membranes: What was the evidence in this work?

In the field of cosmetics: Why focus on cosmetics? Is it related to this work's aim?

Introduction

chitosan stearate (MCh), citric acid (CA), salicylic acid (SA) or ginger extract (VE): Briefly explain the earlier studies related to the present study. 

Now the introduction part is too much information, though these details seem relevant to this study, the reader gets tired, so better to be concise.

2. Materials and Methods

Explain the vendor details for all the raw materials used, like Dextran, citric acid (CA), salicylic acid (SA) or ginger extract....

was obtained by the method described by Ivane et al. [32]. : Better to briefly explain the method to readers. 

were mixed and kept at 100 oC: All the components mixed as a powder? or in liquid?

0.2 g lipase was added: Better to explain the reason to add lipase, is it a crosslinker?

modified chitosan was separated by centrifugation: How did you centrifuge, it seems all the raw materials were added as powder. Did you use ethanol to solubilise all the composite materials? Explain this part clearly.

Preparation of films: So authors fabricated films. Mention this in the abstract. 

Line 175 Dextran (Dex): Already abbreviation introduced above. 

 salicylic acid (SA), ginger extract (VE) were prepared:  salicylic acid (SA), and ginger extract (VE) were prepared

2.4.2. Fourier-transform infrared spectroscopy (FTIR): Explain the sample preparation. 

measurements were performed on swollen samples, shaped as plates with a thickness: Why swollen samples? the properties should be checked as fabricated, did you have control films without swollen? explain the method of swelling. 

Following cooling, the absorbance was measured at 660 nm: It is not clear, how the authors measured at 660 nm without chromogen. How this measures the antiinflammatory of samples? If possible, try to follow some other method to measure the anti-inflammatory effect. 

3. Results and discussions

Figure 1. y axis is missing. label the axis. Why FTIR for two samples? what about the other samples listed in Table 1. 

Figure 2: Change the pattern of graphs in C and D. Add p value and significance by a symbol. Follow this comment for the rest of the graph diagram in other Figs.

Table 1. Average pore sizes of composite biomaterials evaluated from SEM micrographs by ImageJ 376 software: Its Table 2

Figures 5, 6, 7,8,9: Follow the earlier comments.

Try to combine the bar Figures and reduce the number of Figs.

5. Conclusions: Concise.

Reviewer 2 Report

The manuscript entitled “Dextran-Chitosan Composites: Biological, Antioxidant, and 2 Anti-Inflammatory Properties” describes the preparations of composites with matrices comprised of  dextran (Dex) and chitosan (Ch), with fillers such as chitosan stearate (MCh), citric acid, salicylic acid, or ginger extract.  The biological and inflammatory of these composites was investigated. The authors report that the addition of modified chitosan into the Dex:Ch samples yielded a stiff and dense matrix with a 17-fold higher elastic modulus than that of the Dex:Ch matrix which lacked this additive. It was also reported that the addition of citric acid and ginger extract enhanced the anti-oxidative properties of the matrix. Lipid peroxidation inhibitory assay studies revealed that the best inhibition performance (83%) was achieved when the matrix with a composition of Dex:Ch:MCh:citric acid.

            Overall, the research is well-executed and the reported results are well-supported by the provided data. The research is of interest both from a scientific perspective and from an applied standpoint based on the antioxidative properties of these materials. Overall, I believe that this manuscript is suitable for publication pending minor revisions, such as those suggested below.

Lines 17 and 18: Error margins may be needed for reported numerical values such as the compressive nominal stress and compression percentages.

Line 21: error margins may be needed for the inhibition value.

Lines 29-34: A reference should be provided for the definition and description of polysaccharides.

Lines 35-40: A reference should be provided for the statement indicating that natural polysaccharides can be obtained from sources such as seaweed, starch, and cellulose and possibly also for the statement indicating that polysaccharides are biocompatible.

Line 56: “as to non-toxicity” can be changed to “as to its non-toxicity”.

Lines 56-57: A reference may be needed for thee statement “Chitosan is the second largest biological resource”.

Line 124: “have a wide attention” can be changed to “have drawn wide attention”.

Line 133: The phrase “natural polymers presents o high interest,” is unclear.

Line 148: “mechanical testes” should be changed to “mechanical tests”.

Line 154: “1-buthyl-3-methyl-imidazolium,” should be changed to “1-Buthyl-3-methyl-imidazolium,” as it is at the beginning of this sentence.

Line 168: “7g 1-buthyl-3-methyl-imidazolium chloride” should ppossibluy be changed to “7 g 1-buthyl-3-methyl-imidazolium chloride” or “7 g of 1-buthyl-3-methyl-imidazolium chloride”.

Line 184: “detectorand” should be changed to “detector and”.

Line 207: “10mg” can be changed to “10 mg”.

Line 234: “abilities was determined” can be changed to “abilities were determined”.

Line 250: “resulted mixture” can be changed to “resulting mixture” or “resultant mixture”.

Lines 321-322, Figure 2c and d: The error bars may be a difficult to see. Possibly thicker lines can be used to enhance their visibility.

Lines 326-327, Figure 3c and d: The error bars may be a difficult to see. Possibly thicker lines can be used to enhance their visibility.

Lines 336-339: Error margins may be needed for the compression and nominal stress values.

Lines 356-358: Error margins may be needed for the compression and compressive nominal stress values.

Line 366: Error margins may be needed for the compression value.

Lines 394-395 and 402: Error margins may be needed for the antioxidant capability values.

Line 405: Error margins may be needed for the ABTS radical inhibition value.

Line 411: “As result” can be changed to “As a result”.

Lines 417 and 418: Error margins may be needed for the scavenging activity percentages.

Lines 421-422, Figure 7: Error bars may be needed for the graph in Figure 7.

Line 432: “in a considerably increase” can be changed to “in a considerable increase”.

Lines 443: “that he” should be changed to “that the”.

Lines 456, 460, 461 and 462: Error margins may be needed for the numerical values.

Lines 470-474, 477-478: Error margins may be needed for the compression and compressive nominal stress values.

Lines 481, 482, and 487: Error margins may be needed for the numerical values.

Round 2

Reviewer 1 Report

Minor comments:

Figure 1.: Transmittance unit is missing in axis is it %? and numbers in y axis is missing. 

Combine the Figures 5 to Figure 9 in one or two figures with A, B, C and D and explain the figure legends.

Figure 8: X and Y axis lines missing. Axis labelling is invisible. 

The conclusion section was not shortened. Concise this section. 
